# Spanish Cross-Cultural Adaptation and Validation of the Oslo Sports Trauma Research Centre (OSTRC) Overuse Injury Questionnaire in Handball Players

**DOI:** 10.3390/healthcare11060912

**Published:** 2023-03-21

**Authors:** Jesús Martínez-Cal, Guadalupe Molina-Torres, Elio Carrasco-Vega, Luca Barni, María Isabel Ventura-Miranda, Manuel Gonzalez-Sanchez

**Affiliations:** 1Department of Nursing, Physiotherapy and Medicine, Faculty of Health Sciences, University of Almería, 04120 Almería, Spain; 2Department of Physiotherapy, University of Malaga, 29071 Malaga, Spain; 3Institute of Biomedical Research of Malaga (IBIMA), 29010 Malaga, Spain

**Keywords:** OSTRC, overuse injury, cross-cultural adaptation, validation, questionnaire

## Abstract

Objectives: The aim of this study was the cultural adaptation, Spanish translation and validation of the Oslo Sports Trauma Research Centre (OSTRC) Overuse Injury Questionnaire in an adult population. Design: In this study, a cross-sectional design was used. Methods: This study was divided into two phases: (1) cross-cultural adaptation of the original version of the OSTRC to a Spanish version (OSTRC-Sp) and (2) analysis of the psychometric properties of the OSTRC-Sp. A total of 427 handball players of both sexes and over 18 years of age participated in the study. Results: The translated version of the questionnaire showed a very high internal consistency (Cronbach’s α = 0.954), while the subscales showed an internal consistency between 0.832 and 0.961, with the endmost values being for shoulder and low back pain, respectively. On the other hand, when analysing item responses, the OSTRC-Sp showed ICC values ranging from 0.844 to 0.956, the former being for the first back question (back_1), and the latter for the fourth shoulder question (shoulder_4), in line with most published versions. Conclusion: The Spanish version of the OSTRC is a reliable and valid tool that can be used by researchers and clinicians in a Spanish-speaking population with musculoskeletal disorders.

## 1. Introduction

In recent years, interest in handball has grown at an international level [1], with higher game intensity, speed and scores [2], all of which affect the profile of sporting injuries [3]. A large proportion of injuries in this sport affect the lower (40–69%) and upper extremities (17–40%), followed by the head/face (4–32%) and trunk (2–17%), with the most common injuries occurring in the shoulders (19%), lower back (17%) and knees (16%) [4].

Patient-Reported Outcome Measures (PROMs) have experienced an exponential increase in their use, both in clinical settings and in research. They are tools that evaluate subjective aspects of the patient, related to the level of alteration caused by a pathology, negatively determining aspects such as pain, quality of life, disability, etc. Thanks to PROMs, these subjective variables can be evaluated and monitored, allowing for a simple interpretation of the changes that occur in function, symptomatology and/or different capacities [5]. In addition, the use of scales and questionnaires to assess aspects of functionality in athletes has recently become widespread [6,7]. These tools include the Oslo Sports Trauma Research Centre (OSTRC) Overuse Injury Questionnaire [8]. Despite the presence of OSTRC versions for sports injuries in other languages [8,9,10,11,12], no validation of the OSTRC in Spanish, one of the official languages of the UN and the second most spoken native language in the world, has been found in the adult population [13,14]. Although there is a validated version in Spanish in an adolescent population (12–18 years old), it is not applicable to adults [15], since the differences in the characteristics of the two populations requires a specific validation for adults. Consequently, the aim of this study was to carry out a cross-cultural adaptation of the OSTRC and an analysis of the psychometric properties of the OSTRC-Sp. The OSTRC-Sp is a specific questionnaire in Spanish for overuse injuries that occur in the knee, low back and shoulder, of special relevance in the sports field.

## 2. Methods

### 2.1. Study Design

This study used a cross-sectional design that consisted of two phases: (1) cross-cultural adaptation of the original version of the OSTRC to a Spanish version (OSTRC-Sp); (2) analysis of the psychometric properties of the OSTRC-Sp.

### 2.2. Participants

The sample of participants consisted of handball players from Spain, from different federations, clubs and levels. The inclusion criteria were: (1) Spanish speakers over 18 years of age, (2) both sexes and (3) more than one year of sport practice. On the other hand, the study excluded those participants who did not answer all the questions in all the questionnaires.

### 2.3. Ethical Considerations

This study was conducted in accordance with the ethical principles for medical research involving human subjects, and the data were used in compliance with Organic Law 3/2018 on Data Protection. The Ethics Committee approved the conduct of the study, with protocol number EFM 88/2020. All ethical considerations of the Declaration of Helsinki were always taken into account, including the signing of an informed consent form.

#### The Oslo Sports Trauma Research Centre (OSTRC) Overuse Injury Questionnaire

The OSTRC questionnaire consists of 12 questions on pain, limitation in training and competition and reduction in training volume and performance capacity in sports and allows the assessor to calculate the prevalence and a severity score of overuse injuries affecting a particular anatomical area, being specific to the sporting domain of knee, lower back and shoulder problems. Each body area encompasses 4 questions scored from 0 to 25, whose scores are summed to calculate a severity score from 0 to 100 for each overuse problem, where 0 represents no problem, and 25 represents the maximum level for each question. The values for the intermediate responses were chosen to maintain as even a distribution from 0 to 25 as possible while still using whole numbers. Thus, questions 1 and 4 are scored 0–8–17–25, and questions 2 and 3 are scored 0–6–13–19–25 [8].

### 2.4. Translation and Cross-Cultural Adaptation

The cross-cultural adaptation of the OSTRC to the OSTRC-Sp was carried out following the recommendations of the literature [16,17]. There are five stages into which this process can be divided. Initially, two independent translators, native Spanish speakers, each blinded to the other, translated the original version of the OSTRC into Spanish. The two translations were then analysed to compare them and to be able to identify eventual differences in the translation. The intention was always to maintain the original terminology and meaning of the questionnaire. Minor discrepancies were identified, and a third translator helped to obtain an agreed version of a first draft of the OSTRC-SP. A back-translation into English was then requested to analyze whether the Spanish version of the OSTRC maintained the meaning and objective of each question asked. Some minor discrepancies identified in the translation were discussed, agreed upon and corrected. Once the final version of the OSTRC-Sp was available, a pilot study was carried out with 20 subjects to analyse some of the questions that could create comprehension and response problems. After the analysis of the results obtained in the pilot test, the OSTRC was used to collect a sufficient sample to carry out the data collection necessary for the psychometric analysis of the OSTRC-Sp. The responses to the questionnaires were collected online through Google forms. Figure 1 presents an outline of the steps followed to conduct the cross-cultural adaptation of the OSTRC-Sp.

### 2.5. Questionnaires Used for Construct Validity

#### 2.5.1. Disability of the Arm, Shoulder and Hand (DASH)

The DASH questionnaire is a self-administered instrument, consisting of 30 items and 2 optional modules, with 4 items each, designed to measure the impact of an upper limb injury when playing musical instruments and when performing sport or work [18], although only the optional sport module was selected for the present study. The DASH is a specific instrument for measuring the quality of life related to upper limb problems.

#### 2.5.2. Knee Injury and Osteoarthritis Outcome Score (KOOS)

The KOOS is a questionnaire composed of 42 questions that assesses five subscales: pain, sport and recreational function, activities of daily living, knee-related quality of life and other symptoms. The KOOS has already been validated in Spanish [19]. The internal consistency and reliability of the Spanish version shows Cronbach’s alpha coefficient values of 0.78–0.93 and intraclass correlation coefficient (ICC) between 0.76 and 0.91.

#### 2.5.3. Upper Limb Functional Index (ULFI), Lower Limb Functional Index (LLFI), Spine Functional Index (SFI)

The ULFI, LLFI and SFI are three self-administered questionnaires used as a means of assessing activity and participation limitation due to musculoskeletal disorders of the upper extremities, lower extremities and back, respectively. The Spanish version of the ULFI [20] has an internal consistency corresponding to α = 0.94 and a reliability of r = 0.93. On the other hand, the Spanish version of the LLFI [21] shows an internal consistency and reliability corresponding to Cronbach’s alpha = 0.91 and ICC = 0.96, respectively. Finally, the values of internal consistency and reliability of the Spanish version of the SFI [22] are α = 0.85 and r = 0.96, respectively.

#### 2.5.4. Roland–Morris Disability Questionnaire (RMDQ)

The RMDQ is a 24-item back-specific scale derived from the Disease Impact Profile [23] by adding the phrase “because of my back”. Each item is answered “yes” (1 point) or “no” (0 points). This results in a final score between 0 (no disability) and 24 (maximum disability). The reliability of the Spanish version [24] is reported at ICC = 0.87.

#### 2.5.5. EuroQol_5D-5L/EuroQol_VAS

The EQ-5D-5L is a widely used six-item, non-disease-specific questionnaire [25]. The EQ-5D visual analogue scale (VAS) is used to reflect the respondent’s self-rated health status on a 100 mm scale and is rated from “Best imaginable” (100) to “Worst imaginable” (0). The EQ-5D has been shown to be valid and reliable in the Spanish population [26], presenting an internal consistency with a Cronbach’s alpha = 0.53.

### 2.6. Data Analysis

A descriptive analysis was performed in which the mean and standard deviation of the socio-demographic variables and of all the assessment tools used were calculated. The Kolmogorov–Smirnov test was used to analyse the distribution and normality of the sample. Cronbach’s α coefficients were calculated to analyse the internal consistency of the measures, and the values were classified according to the following scale: Cronbach’s α ≤ 0.40 (poor), 0.60 > Cronbach’s α > 0.40 (moderate), 0.80 > Cronbach’s α ≥ 0.60 (good), Cronbach’s α ≥ 0.80 (excellent) [27]. A gender comparison was also conducted to analyse the possible differences in the items. The Student’s t-test was used to compare parametric variables, and the Wilcoxon test was used for non-parametric variables.

The structure and validity of the construct were analysed on the basis of maximum likelihood extraction (MLE), which entailed the fulfilment of these three requirements: accounting for >10% of variance, Eigenvalue > 1.0 and scree plot inflection point. In addition, to perform the MLE, the requirement of a minimum of 10 subjects per item was satisfied (minimum number = 125; subjects measured = 193) [28].

The formula $SEM=s\sqrt{1-r}$ was used to calculate the standard error of measurement (SEM), (“s” is the test scores’ standard deviation (SD) for both measurements (time 1 and 2), and r is the reliability coefficient for the test and Pearson’s correlation coefficient between test and retest values).

Following the analysis described by Stratford [29], the minimal detectable change 90 (MDC90) was used to measure the sensitivity or measurement error. The formula used for the calculation of the MDC90 was as follows: MDC90=SEM×√2×1.65.

Criterion validity was calculated by analysing the degree of correlation between the OSTRC scale and the Spanish versions of the questionnaires DASH [18], ULFI [20], KOOS [19], LLFI [21], SFI [22], RMDQ [24] and EuroQol [26]. The Pearson’s correlation coefficient was structured according to the following scale: r ≤ 0.49 (poor), 0.50 ≤ r ≤ 0.74 (moderate), r ≥ 0.75 (strong) [30]. SPSS (V.23.0) was used for the statistical analysis.

## 3. Results

Appendix A presents the translated and adapted version of the OSTRC (OSTRC-Sp). Table 1 shows the descriptive characteristics of the sample for those variables whose description is based on the frequency of the response, indicating the frequency of occurrence, the percentage it represents with respect to the total and the cumulative percentage of variables such as sex, the existence of previous injuries or the region affected by the last injury, among others.

Table 2 presents the descriptive characteristics of the variables indicating the minimum, maximum, mean and standard deviation. Variables such as mean age, years since the last injury, as well as all the assessment instruments used in this study and their different sub-scales are included. The total sample used for the development of the study (427 participants) is also presented. The results observed when evaluating the ceiling and floor effect in the OSTRC-Sp showed that 27 participants reached the maximum score, and 9 participants obtained the minimum score, that is, 0.06% and 0.02% of the sample, respectively. Based on these results, the ceiling/floor effect in the OSTRC-Sp could be considered not relevant.

The internal consistency of the OSTRC-Sp as a whole was 0.954, while the subscales ranged from 0.832 (shoulder) to 0.961 (low back pain). On the other hand, when analysing the reliability of the item response, ICC values ranging from 0.844 (back_1) to 0.956 (shoulder_4) were observed. When analysing the SEM results, a value of 2.706 was observed. In turn, the MDC90 result was 6.31.

The construct validity was determined through the maximum likelihood method. The sampling adequacy measure (Kaiser–Meyer–Olkin test) presented a value of 0.756, while the Bartlett’s sphericity test presented a value of 4326.781, with 66 degrees of freedom and a significance of *p* ≥ 0.001. The OSTRC-Sp presents a three-factor solution as in the original version [8], with three factors explaining more than 10% of the variance: 33.404%, 28.474% and 17.703% (Table 3). The sedimentation graph for each of the factors is presented in Figure 2, with an inflection point observed from item 3 onwards.

In addition, the questions referring to the shoulder have a much more balanced loading on each of the factors (Table 4). However, Table 5 shows how each of the items that make up the OSTRC-Sp loads on each of the factors extracted. There are some items where a clear loading on each of the factors can be observed; for example, all the questions referring to the knee load on Factor 2, while all the questions referring to the back load preferentially on Factor 1. 

Table 5 presents the correlations between the OSTRC-Sp and its subscales with the different instruments used to analyse the criterion validity. As can be observed, the subscales of the OSTRC-Sp showed a poor correlation between them, although, when analysed with the total value of the questionnaire, two subscales correlated moderately and strongly, namely, shoulder (r = 0.599) and back (r = 0.784), respectively.

On the other hand, when analysing the correlation with the rest of the instruments, it was observed that each questionnaire showed a moderate correlation with each subscale linked to the segment analysed, i.e., KOOS with OSTRC-SP_knee_, SFI and RMQ with OSTRC-SP_back_ and ULFI and DASH with OSTRC-SP_shoulder_. However, this correlation was stronger with the total value of OSTRC-Sp for all scales. The correlations between each of the instruments used in this study are presented in detail in Table 5.

## 4. Discussion

### 4.1. Translation and Cross-Cultural Adaptation of the OSTRC-Sp

The Spanish version of the OSTRC was translated and adapted cross-culturally following the recommendations of the literature [16] to guarantee equivalence between the original version [8] and the Spanish version, which means that the results obtained will be comparable, both in a clinical and in a research setting, with those obtained with the original version [8] and with other previously published versions, such as the German [11], Brazilian [31], Danish [10], Spanish for the youth population [15] and Japanese [32].

### 4.2. Reliability and Measurement Error

The OSTRC-Sp demonstrated internal consistency, with a Cronbach’s α value of 0.954, while the subscales showed internal consistency values between 0.832 and 0.961, these endmost values being obtained for shoulder and low back pain, respectively. These results are in line with the internal consistency reported for other versions of the OSTRC, e.g., the German one, with a Cronbach’s α of 0.92 [11], the Brazilian (0.93) [31], the Japanese (0.93–0.97) [32], the Spanish for youth population (0.88–0.93) [15] or the Danish (0.80–0.93) [10].

On the other hand, in the analysis of the response to each item, in the OSTRC-Sp, ICC values ranging between 0.844 and 0.956 were observed for the first question on the back (back_1) and the fourth question on the shoulder (shoulder_4), respectively. These values are higher than those observed with the Danish version of the OSTRC, where an ICC = 0.62 [10] was observed, which is slightly higher than that obtained with the Spanish version for the youth population (ICC: 0.85–0.87) [15], but similar to those of other published versions, such as the Japanese (0.89–0.97) [32], Brazilian (0.96) [31] or German (0.91) [11].

The OSTRC-Sp showed an MDC of 6.31, while the SEM had a value of 2.706. These outcome variables have only been studied by two of the published versions: the Spanish one for the youth population [15] (SEM = 1.25–1.90) and the Brazilian one [31] (SEM = 1.14). The minimum detectable change was comparable to that reported for the Spanish version for the youth population (5.31–5.99) [15].

### 4.3. Construct Validity

It was identified that three factors met the criteria used for factor extraction: scree plot inflection point (Figure 2), eigenvalue >1.0 and accounting for >10% of variance (Table 3). This three-factor structure is consistent with the original version of the OSTRC [8]. However, for no other version the construct validity was analysed, thus it would be interesting to design studies to analyse this psychometric variable.

### 4.4. Criterion Validity

To carry out the criterion validity test of the OSTRC-Sp, instruments were used to analyse different body regions, i.e., upper limb, DASH and ULFI; lower limb, KOOS and LLFI; back, SFI and RMQ, in addition to the EuroQol-5D and EuroQol-VAS, which are aimed at assessing the patient’s quality of life. The OSTRC has three specific dimensions for each of the body regions identified. In this sense, it can be observed that the level of correlation of the total score of the OSTRC-Sp was always higher than that observed for each of the subscales (Table 5). It can also be observed that the correlation of the sub-scales was always higher in the sub-scales related to the scale oriented to assess the same body region (Table 5).

It was not possible to compare the results obtained for the criterion validity of the OSTRC-Sp, as none of the published versions carried out the analysis of this psychometric variable, except for the Brazilian version [31]. However, the comparison with the Brazilian version [31] could not be performed, since the tools used for this analysis (Global Perceived Effect scale (GPE) and Numeric Pain Rating Scale (NPRS)) do not coincide with those used for the Spanish version, and neither does the outcome variable that is intended to be correlated between the two tools. On the other hand, the Brazilian version is the only one to have carried out an analysis of the floor effect, that was identified as 60% [31]. These values are not comparable to those observed in the OSTRC-Sp, which presented ceiling/floor effect percentages of 0.06% and 0.02%, respectively. It is likely that the selection of the sample used for the validation of the different versions of the OSTRC may account for this difference.

### 4.5. Implications for Future Research and Clinical Use

For good injury management, it is important to have a tool to monitor musculoskeletal disorders in order to measure their incidence, prevalence and severity over time. In this sense, the tools to be used must be developed using best practice, evidence-based results, reliable, concise and valid data. With this objective in mind, the OSTRC was developed, the different versions of which have allowed its use in different population groups. The OSTRC-Sp has proved to be reliable and valid, and the development strategy followed the recommendations of the literature, thus allowing for the assessment and follow-up of patients with musculoskeletal disorders in the Spanish-speaking population.

However, during the development of this study, it was identified that some versions of the OSTRC have not carried out the analysis of some basic psychometric characteristics that would allow understanding the tool in its entirety, such as criterion validity, construct validity and measurement error. In this sense, it would be necessary to design future studies in order to complete the validation process of these tools.

### 4.6. Strengths and Weaknesses

The present study enabled the cross-cultural adaptation of the OSTRC to a Spanish version and its validation, being Spanish the second most spoken language in the world. Moreover, the cross-cultural adaptation and validation was carried out considering the item/participant ratio recommended by the literature [28]. In this sense, the OSTRC is a questionnaire composed of 12 questions, thus the minimum number of participants required would have been 120. The validation of the OSTRC-Sp was carried out with 427 participants, which is more than three times the recommended number.

However, our study has also some weaknesses that need to be taken into account. In this sense, this study did not analyse any longitudinal psychometric variables, such as sensitivity to change or responsiveness; thus, future studies should be designed and developed to assess these psychometric characteristics in the OSTRC-Sp.

## 5. Conclusions

The Spanish version of the OSTRC is a reliable and valid tool that can be used by researchers and clinicians in a Spanish-speaking population with musculoskeletal disorders. This questionnaire covers different body regions and was validated both in its total value and in each of its sub-categories; thus, it could be used independently for each sub-category if the aim is to evaluate a specific body region.

## Figures and Tables

**Figure 1 healthcare-11-00912-f001:**
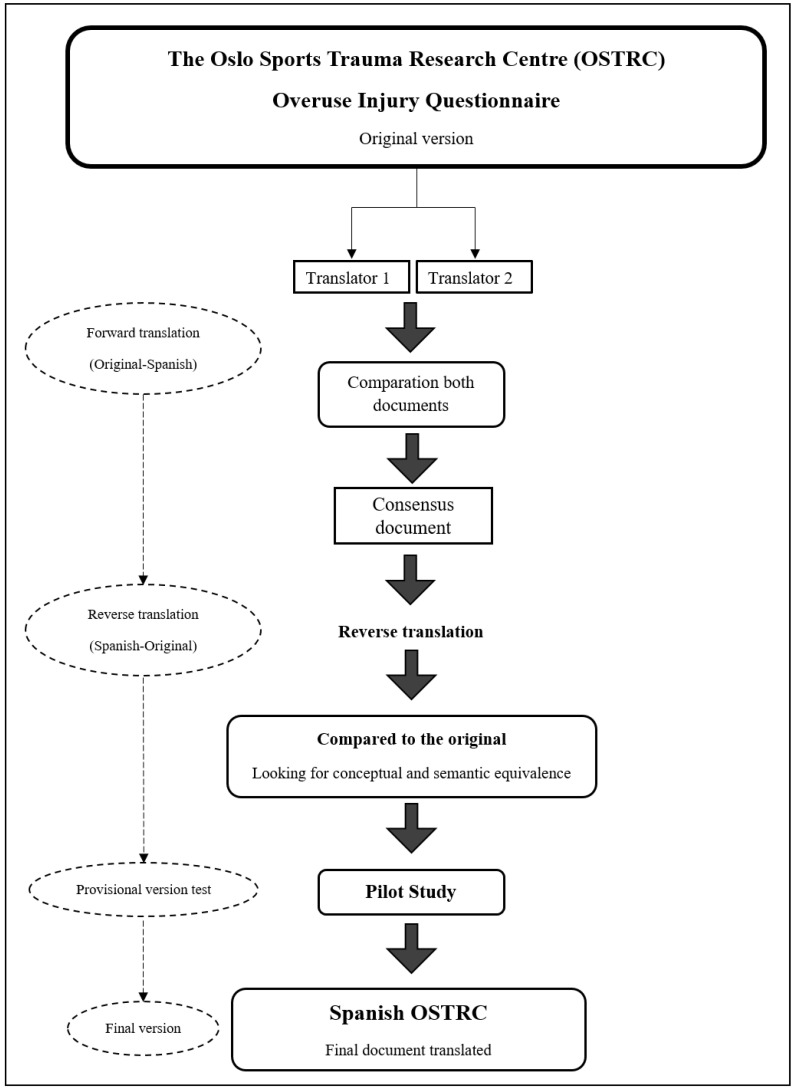
Flowchart of the development process of the OSTRC Spanish version.

**Figure 2 healthcare-11-00912-f002:**
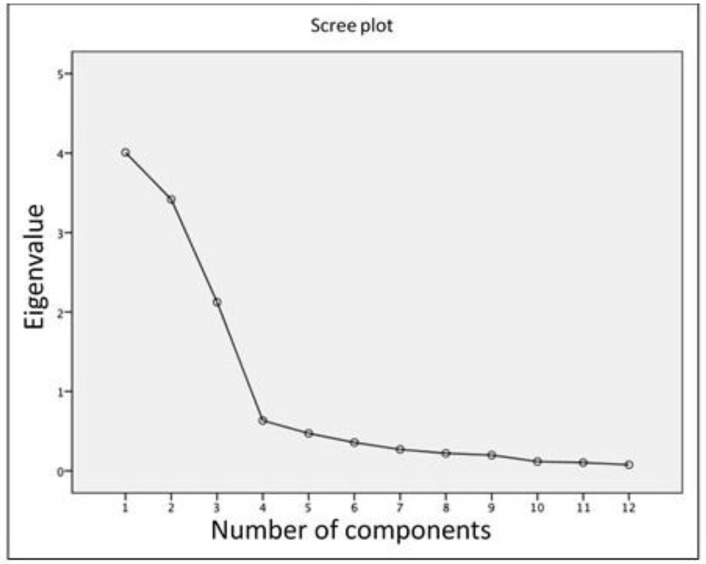
Scree plot of each of the OSTRC-Sp components.

**Table 1 healthcare-11-00912-t001:** Descriptive characteristics of the sample in variables that can be analysed according to the frequency of appearance.

		Frequency	Percentage	Accumulated Percentage
Gender	Man	146	34.2	34.2
	Woman	281	65.8	100
Study level	Basic Education	12	2.8	2.8
	Highschool	156	36.5	39.3
	University studies	211	49.4	88.8
	Postgraduate	48	11.2	100
Civil Status	Single	325	75.9	75.9
	With partner	21	4.9	81.0
	Domestic Partner	21	4.9	85.9
	Married	60	14.1	100
Laterality	Right	369	86.4	86.2
	Left	49	11.5	97.9
	Bilaterality	9	2.1	100
Previous Injuries	No Yes	82 345	19.2 80.8	19.2 100
Region	Shoulder	61	17.9	17.9
	Hand	17	5.0	22.9
	Back	9	2.6	25.5
	Hip	4	1.2	26.7
	Thigh	54	15.8	42.5
	Knee	124	36.4	78.9
	Ankle	58	17.0	95.9
	Foot	14	4.1	100

**Table 2 healthcare-11-00912-t002:** Age and descriptive results of all the outcome variables used in this study.

		Mean ± SD (Min–Max)
Age		25.50 ± 8.456 (18–46)
Years since last injury		4.62 ± 4.384 (0–20)
OSTRC	Knee	5.14 ± 1.987 (4–12)
	Low Back	5.80 ± 3.361 (4–18)
	Shoulder	4.87 ± 8.818 (4–13)
	Total	15.81 ± 4.706 (4–33)
DASH	Disability	51.29 ± 29.686 (12–100)
	Sport	49.14 ± 30.819 (0–100)
Upper Limb Functional Index		48.19 ± 30.469 (0–100)
KOOS	Pain	17.66 ± 10.234 (0–35)
	Symptoms	13.67 ± 8.093 (0–27)
	ADL_Function	33.40 ± 20.238 (0–67)
	Sport	9.65 ± 5.787 (0–19)
	Quality of Life	7.20 ± 4.772 (0–15)
	Total	81.58 ± 49.078 (0–163)
Lower Limb Functional Index		51.59 ± 29.306 (0–100)
Spine Functional Index		48.19 ± 30.402 (0–100)
Roland Morris Questionnaire		11.43 ± 6.839 (0–24)
EuroQol_5D		0.641 ± 0.242 (0.24–1.00)
EuroQol_VAS		47.35 ± 31.158 (0–100)
N		427

SD: standard deviation; OSTRC: Oslo Sports Trauma Research Centre; DASH: Disability of the Arm, Shoulder and Hand; KOOS: Knee Injury and Osteoarthritis Outcome Score.

**Table 3 healthcare-11-00912-t003:** Results of the explained variance analysis of each of the items in which the OSTRC-Sp is structured.

Component	Initial Eigenvalues			Sums of Extraction of Charges Squared		
	Total	% de Variance	% Accumulated	Total	% de Variance	% Accumulated
1	4.009	33.404	33.404	4.009	33.404	33.404
2	3.417	28.474	61.878	3.417	28.474	61.878
3	2.124	17.703	79.582	2.124	17.703	79.582
4	0.634	5.280	84.862			
5	0.472	3.936	88.799			
6	0.357	2.971	91.770			
7	0.270	2.249	94.019			
8	0.221	1.843	95.862			
9	0.198	1.651	97.513			
10	0.117	0.977	98.490			
11	0.104	0.868	99.358			
12	0.077	0.642	100.000			

**Table 4 healthcare-11-00912-t004:** Matrix of the OSTRC-Sp components, where you can see their distribution between each of the factors of the OSTRC-Sp.

		Component		
		1	2	3
OSTRC_1. Knee	Have you had any difficulties participating in normal training and competition due to knee problems during the past week?	0.527	0.731	0.153
OSTRC_2. Knee	To what extent have you reduced you training volume due to knee problems during the past week?	0.479	0.792	0.174
OSTRC_3. Knee	To what extent have knee problems affected your performance during the past week?	0.554	0.759	0.169
OSTRC_4. Knee	To what extent have you experienced knee pain related to your sport during the past week?	0.411	0.810	0.138
OSTRC_1. Lower Back	Have you had any difficulties participating in normal training and competition due to lower back problems during the past week?	0.624	−0.231	−0.576
OSTRC_2. Lower Back	To what extent have you reduced you training volume due to lower back problems during the past week?	0.668	−0.271	−0.428
OSTRC_3. Lower Back	To what extent have lower back problems affected your performance during the past week?	0.674	−0.201	−0.556
OSTRC_4. Lower Back	To what extent have you experienced lower back pain related to your sport during the past week?	0.709	−0.050	−0.490
OSTRC_1. Shoulder	Have you had any difficulties participating in normal training and competition due to shoulder problems during the past week?	0.566	−0.412	0.457
OSTRC_2. Shoulder	To what extent have you reduced you training volume due to shoulder problems during the past week?	0.577	−0.497	0.315
OSTRC_3. Shoulder	To what extent have shoulder problems affected your performance during the past week?	0.565	−0.496	0.551
OSTRC_4. Shoulder	To what extent have you experienced shoulder pain related to your sport during the past week?	0.510	−0.435	0.589

**Table 5 healthcare-11-00912-t005:** Correlation matrix between the total OSTRC-Sp score and all the reference instruments used for criterion validity, as well as their subscales.

		OSTRC			
		Knee	Back	Shoulder	Total
OSTRC	Knee_Total	1	0.016	0.222 **	0.405 **
	Back_Total	0.016	1	0.223 **	0.784 **
	Shoulder_Total	0.222 **	0.223 **	1	0.599 **
	Total	0.405 **	0.784 **	0.599 **	1
Lower Limb Functional Index		0.581 **	0.398 **	0.490 **	0.866 **
KOOS	Pain	0.595 **	0.397 **	0.504 **	0.881 **
	Symptoms	0.584 **	0.401 **	0.493 **	0.870 **
	ADL_Function	0.585 **	0.399 **	0.496 **	0.872 **
	Sport	0.585 **	0.398 **	0.491 **	0.868 **
	QoL	0.593 **	0.400 **	0.497 **	0.875 **
	Total	0.588 **	0.400 **	0.497 **	0.874 **
Spine Functional Index		0.405 **	0.595 **	0.495 **	0.877 **
Roland-Morris Questionnaire		−0.281 **	−0.565 **	−0.363 **	−0.665 **
Upper Limb Functional Index		0.401 **	0.491 **	0.584 **	0.867 **
DASH	Disability	0.403 **	0.493 **	0.592 **	0.870 **
	DASH_Sport	0.396 **	0.482 **	0.604 **	0.870 **
EuroQoL	EuroQoL_5D	−0.337 **	−0.525 **	−0.401 **	−0.699 **
	EuroQoL_VAS	0.361 **	0.543 **	0.424 **	0.787 **

**: Significance: *p* ≥ 0.01 (bilateral).

## Data Availability

Data is contained within the article or Appendix A.

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
