# Peer review of "Spanish Cross-Cultural Adaptation and Validation of the Oslo Sports Trauma Research Centre (OSTRC) Overuse Injury Questionnaire in Handball Players"

_healthcare, 2023, doi:10.3390/healthcare11060912_

Round 1
Reviewer 1 Report
Similarity rate should be 15%
Introduction
6, 13 references not found.
OSTRC-Sp :should be explained in the first written
Methods
2.4 Translation and cross-cultural adaptation
Can the following article be used instead of 16 references?
(Hernandez A, Hidalgo MD, Hambleton RK, Gómez-Benito J. International Test Commission guidelines for test adaptation: A criterion checklist. Psicothema. 2020 Aug;32(3):390-398. doi: 10.7334/psicothema2019.306.)
Figure 1. Flowchart of de development process OSTRC Spanish version. (instead of ‘de’, it should be ‘the’)
2.5.2. Knee Injury and Osteoarthritis Outcome Score (KOOS)
What is the meaning of ICC ?
2.5.5. EuroQol_5D / EuroQol_VAS
EuroQol 5D, The EQ-5D expansion should be written
3. Results
Table 1:
‘Bilaterality’ instead of ‘ambidextrous’
Instead of ‘alteration segment’, ‘region’ should be written.
Table.2
It is better if it is given as mean ± SD (Min-Max)
(for example (Age: 25.5±8.45) (18-46))
low back instead of back
The meanings of the abbreviations should be written under the table. (DASH, KOOS, SD……) (For exaple SD:Standart deviasyon etc…)
Table.5
no need to write the other part in the correlation table
Discussion:
No need for the first sentence of the discussion (Written in introduction)
References
7 references should be corrected
(13 references) can it be in English?
Author Response
We would like to thank the Editor and reviewers for their thoughtful and constructive comments. We have considered all suggestions, and have incorporated them into the revised manuscript. Changes to the original manuscript are identified by highlights (in yellow background). After corrections made, we believe that our document is much easier to read and understand. An itemized point-by-point response to the reviewers’ comments is presented below.
Thank you very much for offering us the possibility of reviewing the document and being able to complement it with the suggestions and comments made by the reviewers. We have followed all the suggestions made by the reviewer to understand that the document evolves positively.
Reviewer: 1
Introduction
6, 13 references not found.
Authors’ answer: Thank you for your suggestion. It is true that the reference number 6 does not appear indexed in pubmed, it belongs to the Spanish journal Fisioterapia from the Elsevier publishing house and can be located at the following link:
https://www.elsevier.es/es-revista-fisioterapia-146-articulo-propiedades-psicometricas-cuestionarios-funcionalidad-poblacion-S0211563820300195
Reference number 13 corresponds to the annual report of the prestigious Cervantes Institute, a world reference as far as the Spanish language is concerned. You can find this report at the following link:
https://cvc.cervantes.es/lengua/espanol_lengua_viva/pdf/espanol_lengua_viva_2018.pdf
OSTRC-Sp: should be explained in the first written
Authors’ answer: Thank you for your comment. We have added a short explanation at the end of the introduction.
Methods
2.4 Translation and cross-cultural adaptation
Can the following article be used instead of 16 references?
(Hernandez A, Hidalgo MD, Hambleton RK, Gómez-Benito J. International Test Commission guidelines for test adaptation: A criterion checklist. Psicothema. 2020 Aug;32(3):390-398. doi: 10.7334/psicothema2019.306.)
Authors’ answer: Thank you for your suggestion. Analyzing both documents we have verified that both references are complementary. The new recommended reference has been added.
Figure 1. Flowchart of de development process OSTRC Spanish version. (instead of ‘de’, it should be ‘the’)
Authors’ answer: Thank you for your comment, we have made the indicated correction.
2.5.2. Knee Injury and Osteoarthritis Outcome Score (KOOS)
What is the meaning of ICC ?
Authors’ answer: Thank you for your comment. The meaning of ICC is intraclass correlation coefficient. We have added this definition in the indicated point.
2.5.5. EuroQol_5D / EuroQol_VAS
EuroQol 5D, The EQ-5D expansion should be written
Authors’ answer: Thank you for your suggestion. We have modified the point indicated so that it is clear that it refers to the expansion of 5 levels and added the corresponding reference.
- Results
Table 1:
‘Bilaterality’ instead of ‘ambidextrous’
Instead of ‘alteration segment’, ‘region’ should be written.
Authors’ answer: Thank you for your comment. Both points have been corrected in the table 1.
Table.2
It is better if it is given as mean ± SD (Min-Max)
(for example (Age: 25.5±8.45) (18-46))
low back instead of back
The meanings of the abbreviations should be written under the table. (DASH, KOOS, SD……) (For exaple SD:Standart deviasyon etc…)
Authors’ answer: Thank you for your suggestion. All the suggestions indicated have been modified or added in the table.
Table.5
no need to write the other part in the correlation table
Authors’ answer: Sorry, but we're not quite sure what exactly the reviewer is referring to.
Discussion:
No need for the first sentence of the discussion (Written in introduction)
Authors’ answer: Thank you for your suggestion. We have removed the indicated sentence.
References
7 references should be corrected
Authors’ answer: Thank you for your suggestion. The indicated reference has been corrected.
(13 references) can it be in English?
Authors’ answer: Thank you for your suggestion. I'm sorry, it is a text entirely in Spanish and there is no reference in English.

Reviewer 2 Report
It is my pleasure to have the opportunity to review this manuscript. This manuscript is a report of an attempt to translate the OSTRC into Spanish as "Spanish cross-cultural adaptation and validation of the Oslo Sports Trauma Research Centre (OSTRC) Overuse Injury Questionnaire in Handball Players.”
I have read the manuscript and would like to comment as follows
1.(Introduction)
The OSTRC citation (Ref. 8) from which the translation is taken is from the first edition in 2013; a revised edition of the OSTRC was published in 2020. If there is a reason to use the 2013 edition, please confirm the reason.
2. (line40-41, Table1, Table2, Table4, Table5)
The formatting does not seem to be consistent. Some parts are in bold, some in portrait, etc. Please check again.
3. (Methods)
How were the responses collected in the pilot study? Many studies use questionnaires or web-based surveys.
4. (Methods)
The document describes the need to modify the language and cultural characteristics when translating from English to Spanish and reverse translation, but specific details need to be added.
5. (Methods)
Please add the reasons why you used the assessment items used for validation (DASH, KOOS, ULFI, etc.). Other languages use different assessment items.
Author Response
We would like to thank the Editor and reviewers for their thoughtful and constructive comments. We have considered all suggestions, and have incorporated them into the revised manuscript. Changes to the original manuscript are identified by highlights (in yellow background). After corrections made, we believe that our document is much easier to read and understand. An itemized point-by-point response to the reviewers’ comments is presented below.
Thank you very much for offering us the possibility of reviewing the document and being able to complement it with the suggestions and comments made by the reviewers. We have followed all the suggestions made by the reviewer to understand that the document evolves positively.
Reviewer: 2
1.(Introduction)
The OSTRC citation (Ref. 8) from which the translation is taken is from the first edition in 2013; a revised edition of the OSTRC was published in 2020. If there is a reason to use the 2013 edition, please confirm the reason.
Authors’ answer: Thank you for your comment. First of all, we must say that within the group of authors there was a deep debate beforehand about which of the two versions to translate into Spanish. For our criteria, the modern version was not the one that best suited our needs, as we wanted to measure a specific thing. Therefore, the patient profile forced us to decide for one of the two. What made us decide for the old version is that it is divided into 3 subcategories (knee, low back and shoulder) and they are independently evaluable, which allows us on the one hand, independent validation of the questionnaire and on the other hand, a better assessment of the patients we are analyzing. This does not mean that in the future we will not consider the validation of the modern version, which at the same time will be even better, because the research group already has a complete overview of the old version of the questionnaire. In fact, there are examples of widely used questionnaires such as the SF-36, which despite having a more modern reduced version such as the SF-12 are still widely used.
- (line40-41, Table1, Table2, Table4, Table5)
The formatting does not seem to be consistent. Some parts are in bold, some in portrait, etc. Please check again.
Authors’ answer: Thank you for your comment. There was no such problem in the original tables. Maybe it is a problem that occurs when switching between different text editors. In any case, it has been rechecked and the errors found have been corrected.
- (Methods)
How were the responses collected in the pilot study? Many studies use questionnaires or web-based surveys.
Authors’ answer: Thank you for your comment. The responses to the questionnaires were collected online through Google forms. This sentence has been added in the methodology (2.4.).
- (Methods)
The document describes the need to modify the language and cultural characteristics when translating from English to Spanish and reverse translation, but specific details need to be added.
Authors’ answer: Thank you for your comment. Specific details have been added in the methodology (2.4.).
- (Methods)
Please add the reasons why you used the assessment items used for validation (DASH, KOOS, ULFI, etc.). Other languages use different assessment items.
Authors’ answer: Thank you for your comment. When the authors were deciding which questionnaires, we should use to carry out the external validity of the OSTRC, we could observe much heterogeneity in the questionnaires used. Given that the subscales presented by the OSTRC evaluate the upper limb, back and lower limb, we decided to reference the most used questionnaires to evaluate said regions. It can be observed in the tables how there is a more significant correlation between the subscales and the different specific questionnaires for each region. We also included the EuroQoL-5d as the questionnaire with the highest frequency of use in assessing people's quality of life.
